# GROUPLANE: END-TO-END 3D LANE DETECTION WITH CHANNEL-WISE GROUPING

## ABSTRACT

Efficiency is quite important for 3D lane detection while previous detectors are either computationally expensive or difficult for optimization. To bridge this gap, we propose a fully convolutional detector named GroupLane, which is simple, fast, and still maintains high detection precision. Specifically, we first propose to split extracted feature into multiple groups along the channel dimension and employ every group to represent a prediction. In this way, GroupLane realizes end-to-end detection like DETR based on pure convolutional neural network. Then, we propose to represent lanes by performing row-wise classification in bird's eye view and devise a set of corresponding detection heads. Compared with existing row-wise classification implementations that only support recognizing vertical lanes, ours can detect both vertical and horizontal ones. Additionally, a matching algorithm named single-win one-to-one matching is developed to associate prediction with labels during training. Evaluated on 3 benchmarks, OpenLane, Once-3DLanes, and OpenLane-Huawei, GroupLane adopting ConvNext-Base as the backbone outperforms the outstanding method PersFormer by 13.6% F1 score in the OpenLane validation set. Besides, GroupLane with ResNet18 still surpasses PersFormer by 4.9% F1 score, while the inference speed is $7\times$ faster.

## 1 INTRODUCTION

Achieving rapid and precise lane detection is critical for realizing robust driver assistance systems and safe autonomous driving (Pan et al., 2018; Li et al., 2019; Grigorescu et al., 2020). Early publications are mostly about 2D lane detection, which recognizes lanes in the 2D camera image plane (Tang et al., 2021). By contrast, recent attention of the research community is paid to 3D lane detection that directly localizes lanes in the 3D world space, because it benefits downstream tasks like planning (Garnett et al., 2019).

Various detectors have been proposed in recent years (Hou et al., 2019; Xu et al., 2020), while most of them rely on complex post-processing operations and troublesome hyper-parameter tuning (Tabelini et al., 2021a). Although there exist a few detectors that do not demand post-processing by resorting to the power of Transformer and representing lanes as parametric curves (Liu et al., 2021b), they suffer from heavy computing burden and are difficult for optimization (Tabelini et al., 2021b). Thus, the performance is still limited and does not satisfy the demand in practical applications.

To address the aforementioned problems, we first propose a method that realizes end-to-end detection like DETR (Carion et al., 2020) but based on only fully convolutional neural network (CNN). Specifically, we split the convolutional channels of a feature map into multiple groups and employ every group to represent a target. During training, the predicted attributes of various channel groups are matched with labels based on one-to-one matching to compute loss, and no post-processing is required for inference. In this way, our designed detector maintains the advantages of CNN and Transformer simultaneously, i.e., computationally efficient and end-to-end.

The following problem is how we model the complex topology structure of 3D lanes. In this work, we propose to perform row-wise classification (Liu et al., 2021a) in the bird's eye view (BEV). Compared with existing 3D lane detectors based on anchor lines or point clusters (Lee et al., 2017), ours is easier for optimization, because it does not require operations like clustering and non-maximum suppression (NMS) (Ren et al., 2015). Besides, 3D lanes in BEV are more visually straight and in parallel to each other, which naturally benefits row-wise classification.

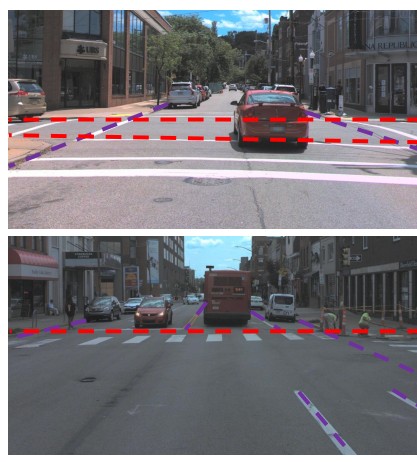

Figure 1: Examples of vertical lanes and horizontal lanes, which are highlighted in purple and red.

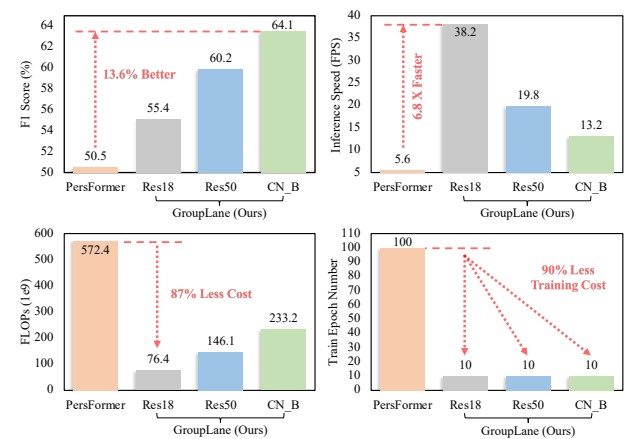

Figure 2: Comparison between PersFormer and GroupLane in OpenLane adopting various backbones (ResNet18, ResNet50, and ConvNext-Base) on different metrics, i.e., F1 score, inference speed, FLOPs, and epoch number.

However, the problem is the conventional implementation of using row-wise classification to represent curves only supports recognizing vertical lanes (Qin et al., 2020), which are marked in purple in Fig. 1. When horizontal lanes (the red ones shown in Fig. 1) appear, the performance becomes poor. To address this issue, we propose to build two groups of heads, the vertical group and horizontal group, which are responsible for recognizing horizontal lanes and vertical lanes, respectively. Additionally, a novel one-to-one matching algorithm named single-win one-to-one matching (SOM) is developed to match predictions with targets during training for computing loss.

Combining the aforementioned techniques, a fully convolutional end-to-end 3D lane detector GroupLane is proposed. Extensive experiments are conducted in the OpenLane (Chen et al., 2022), Once-3DLanes (Yan et al., 2022), and OpenLane-Huawei (OpenLane-V2 Dataset Contributors, 2023) benchmarks to verify its performance. Although simple, we observe that GroupLane surpasses all counterparts by large margins in all the benchmarks. For example, we compare GroupLane with Pers-Former (Chen et al., 2022) using four metrics, i.e., F1 score, inference speed, floating point operations (FLOPs), and training epoch number. The backbone employed by PersFormer is EfficientNet-B7 (Tan & Le, 2019). The results are visualized in Fig. 2. As shown, adopting ConvNext-Base (Liu et al., 2022) as the backbone, GroupLane outperforms PersFormer by 13.6% F1 score. In addition, GroupLane with ResNet18 (He et al., 2016) can still surpass PersFormer by 4.9% F1 score. Notably, the inference speed of GroupLane with ResNet18 is nearly 7× faster than PersFormer and the FLOPs is only 13.3% of it. Meanwhile, GroupLane is highly training economical, because it only uses 10 training epochs while PersFormer demands 100 epochs.

Summarily, our technical contributions include: (i) We develop a strategy that splits feature into multiple groups and employs every group to represent a lane. By matching predictions with targets based on SOM, a fully convolutional detector realizing end-to-end detection like DETR is proposed. (ii) We propose to represent lanes by performing row-wise classification in BEV and design the corresponding heads. These heads can recognize both vertical and horizontal lanes.

## 2 RELATED WORK

**Lane Detection.** As a fundamental perception task in autonomous driving, lane detection is widely studied (Tang et al., 2021). Previous detectors mostly recognize lanes in the 2D camera plane, which is known as 2D lane detection. According to the strategies of modeling lanes, existing detectors can be categorized into 4 classes, i.e., segmentation based (Hou et al., 2019; Lee et al., 2017), anchor based (Tabelini et al., 2021a), parametric curve based (Liu et al., 2021b), and row-wise classification based (Liu et al., 2021a). Since the segmentation based and anchor based methods need complex post-processing like NMS and clustering, they are often slow and sensitive to the tuning of hyper-parameters (Pan et al., 2018; Xu et al., 2020). By contrast, the parametric curve based methods are end-to-end by representing lanes as parametric equations. However, they suffer

from serious computing burden because it is usually based on the Transformer architecture (Tabelini et al., 2021b; Liu et al., 2023). The row-wise classification based detectors are also end-to-end and consume acceptable computing resource. However, their performances are relatively poor in existing 2D lane detectors (Qin et al., 2020). Although our proposed GroupLane is also based on row-wise classification, we extend it to 3D lane detection and achieve much more impressive performance.

**3D Lane Detection.** 2D lane detectors generate results in the 2D camera plane, which is inconsistent with downstream tasks like planning. To overcome this problem, increasing attention is paid to 3D lane detection (Guo et al., 2020; Wang et al., 2022; Luo et al., 2023). Compared with 2D lane detection, there are two new challenges. The first one is how to aggregate features from 2D camera planes to the 3D world space, and various strategies are proposed. For instance, Anchor3DLane (Huang et al., 2023) projects anchor lines back to the camera plane to sample features. CurveFormer (Bai et al., 2022) initializes query tensors in the world space and aggregates information through deformable attention (Zhu et al., 2020). Another challenge is how to model 3D lanes. Existing 3D lane detectors mainly follow the strategies in 2D lane detection. To the best of our knowledge, no existing 3D lane detector employs the row-wise classification strategy adopted by GroupLane.

**BEV Feature Generation.** As mentioned before, the operation of transforming 2D image feature to the 3D world space is important for 3D lane detection (Li et al., 2022b;c;d). A direct idea of performing this transformation is generating BEV feature based on geometric constraints explicitly. There are primarily two strategies for producing the BEV feature, projecting feature to the BEV plane (Philion & Fidler, 2020) and sampling feature from the camera plane (Li et al., 2022a). For the first one, the network splits the BEV plane into uniform grids and encodes the feature from the camera plane into these grids. In the second one, numerous BEV queries are initialized and utilized to aggregate information from the camera plane through deformable attention. GroupLane follows the first strategy because of its simple form.

## 3 METHOD

In this section, the overall framework of GroupLane is introduced in Section 3.1. Then, a description about how the fully convolutional end-to-end detection is realized is presented in Section 3.2. Afterwards, the SOM strategy is explained in Section 3.3. Finally, we explain the design of the detection heads in Section 3.4.

### 3.1 OVERALL FRAMEWORK

Before explaining our contributions, we first describe the overall framework of GroupLane as shown in Fig. 3. In each iteration, a batch of monocular images $I \in \mathbb{R}^{B \times 3 \times H_i \times W_i}$ is input to the backbone (such as ResNet50) for extracting multiple levels of feature maps, where $B$, $H_i$, and $W_i$ denote the batch size, image height, and image width, respectively. Then, a neck (e.g., Second FPN (Yan et al., 2018)) is built to fuse the multiple levels of feature maps as a single feature map $F_s \in \mathbb{R}^{B \times C \times H_s \times W_s}$. After obtaining $F_s$, the LSS module (Philion & Fidler, 2020) is adopted to transform the camera view feature into the BEV feature $F_b \in \mathbb{R}^{B \times C \times H_b \times W_b}$, where $C$, $H_b$, and $W_b$ represent the channel number, height, and width of the BEV feature, respectively.

Subsequently, we split $F_b$ into $2 \times N$ groups of features $\{f_b^i\}_{i=1}^{2N}$ in the channel dimension, where $f_b^i \in \mathbb{R}^{B \times C_g \times H_b \times W_b}$ denotes the $i_{\text{th}}$ group and $2 \times N \times C_g = C$. The first $N$ groups of feature are input to the vertical head group and the other $N$ groups are for the horizontal head group. In this work, we use every group of features to represent a candidate target, which is similar to the usage of queries in the detection Transformer (Carion et al., 2020). In the following network, all the convolutional layers are implemented based on group convolution (Xie et al., 2017). Both the vertical and horizontal head groups comprise 6 heads, which are for predicting the existence confidence, visibility, category, row-wise classification index, x-axis offset, and z-axis offset, respectively.

### 3.2 CHANNEL GROUPING

In this work, we hope $F_b$ can behave like the query tensors in DETR (Carion et al., 2020) to realize end-to-end 3D lane detection. To this end, we first split $F_b$ as multiple groups of features $\{f_b^i\}_{i=1}^{2N}$ in the channel dimension. This process is illustrated in Fig. 4, where various channel groups are marked in different colors. In all the convolution layers after the BEV encoder, we employ group convolution rather than classical convolution. The convolution group number is also set to $2 \times N$. In this way, we

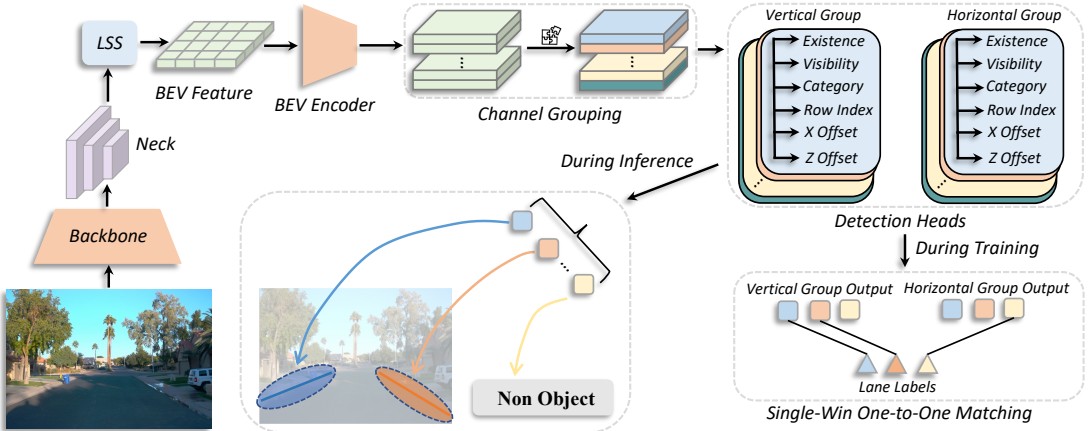

Figure 3: Overall framework of GroupLane. During training, predictions are matched with lane labels based on our proposed SOM strategy to compute loss. During inference, GroupLane generates detection results in an end-to-end fashion without post-processing.

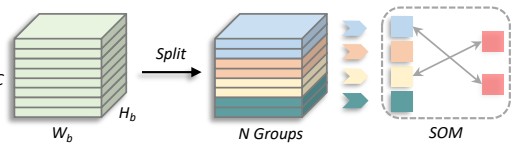

Figure 4: The diagram of how the channel grouping strategy realizes end-to-end 3D lane detection. The feature maps and predictions corresponding to various channel groups are marked in different colors. Lane labels are colorized in salmon. The SOM step only exists in the training process.

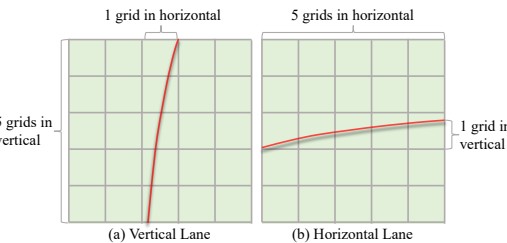

Figure 5: The diagram of vertical lanes and horizontal lanes. We define the lanes crossing more grids in vertical than grids in horizontal as vertical lanes. Conversely, they are horizontal lanes.

ensure that there is no information interaction between different groups of features, and every group concentrates on its own corresponding target.

As shown in Fig. 4, after being processed by group convolutional layers, $2 \times N$ lane detection candidates are generated, and each candidate corresponds to a feature group in $\{f_b^i\}_{i=1}^{2N}$. During the training process, we match the candidates with lane labels using SOM to compute loss. For the inference process, we directly judge which predictions are valid by confidence filtering. In this way, end-to-end detection without post-processing is realized. Besides, the whole process is fully convolutional rather than based on Transformer attention, which saves much computational cost.

### 3.3 SINGLE-WIN ONE-TO-ONE MATCHING

There exist two groups of heads, the vertical and horizontal groups, which are for recognizing vertical and horizontal lanes, respectively. Visualizing in the BEV plane, we define the lanes crossing more grids in vertical than grids in horizontal as vertical lanes, which are illustrated in Fig. 5 (a). Conversely, the lanes crossing more grids in horizontal are horizontal lanes as shown in Fig. 5 (b).

Previous row-wise classification detectors usually assume all lanes are vertical and only build a single group of heads. Differently, in this work, we build the vertical head group that performs row-wise classfication in horizontal rows of grids and this group is for recognizing vertical lanes. Similarly, the horizontal group conducts row-wise classfication in vertical rows of grids and is to detect horizontal lanes. During training, both the vertical and horizontal head groups produce $N$ lane predictions, so there are totally $2N$ output predictions. The problem is how to assign labels to these $2N$ predictions for computing loss.

According to Fig. 5, we can find that a lane can be represented by both the vertical and horizontal head groups, and the difference is the crossed grid number. For example, as shown in Fig. 5 (a), the lane only crosses 1 grid in horizontal while 5 grids in vertical. Therefore, using the vertical head group can model this lane more precisely. During training, the label of this lane should be

assigned to the vertical head group to compute loss. Based on the above insight, we propose the SOM strategy. In this strategy, we first match labels with the $N$ predictions produced by one of the two head groups separately. The matching cost is defined the same as loss functions, which will be described in Section 3.4. In this way, any label is matched with two model predictions, one generated by the vertical head group and the other from the horizontal head group. Afterwards, we compare their crossed grid numbers to determine which prediction to assign.

## 3.4 DETECTION HEADS

In this part, we describe the implementation of our designed two head groups (the vertical and horizontal groups) briefly. Every group consists of 6 heads, i.e., the existence head, row index head, visibility head, category head, x-axis offset head, and z-axis offset head. Each head contains 2 convolutional layers. The input to these heads is the BEV feature $F_b \in \mathbb{R}^{B \times C \times H_b \times W_b}$. We split $F_b$ into $F_b^v \in \mathbb{R}^{B \times C/2 \times H_b \times W_b}$ and $F_b^h \in \mathbb{R}^{B \times C/2 \times H_b \times W_b}$, and they serve as the input to the two head groups, respectively.

**Existence head.** As mentioned in Section 3.2, $2N$ detection candidates are generated by the horizontal and vertical head groups, and not all candidates are valid targets. Some candidates correspond to background regions. The existence head is to predict the confidence that a candidate is a valid lane rather than belonging to background regions and the head output is $y_e \in \mathbb{R}^{B \times N}$.

**Visibility head.** In this work, we split the BEV plane into uniform grids. The shape of these grids is $H_b \times W_b$, which is the same as the BEV feature resolution. Then, for the vertical head group, a set of y-axis lines ($\{y = y_i\}_{i=1}^{H_b}$) are pre-defined in this BEV space (x-axis lines for the horizontal head group). The visibility head is to estimate whether there are visible lanes crossing these y-axis lines. For example, as shown in Fig. 6 (a), the rows marked in red do not have visible lanes. To predict whether a row is crossed by a lane correctly, the output of the visibility head needs to aggregate the information from all grids in this row. To this end, the visibility head applies the max pooling operation to the input feature along the x-axis dimension, so the output is $y_v \in \mathbb{R}^{B \times N \times H_b \times 1}$.

**Row index head.** In GroupLane, we represent 3D lanes with row-wise classification (Qin et al., 2020), which splits a 2D plane into many grids and distinguishes which grid in every row contains lanes. Although there are a few previous detectors adopting this strategy (Yoo et al., 2020), all them conduct row-wise classification in the camera plane, as shown in Fig. 6 (a). In contrast to them, we perform row-wise classification in the BEV plane, which is illustrated as Fig. 6 (b). Comparing Fig. 6 (a) and (b), we can observe that the lanes in the camera plane present more complex topologies. Notably, there is no lane in the top rows of grids in Fig. 6 (a), marked in red. By contrast, all rows

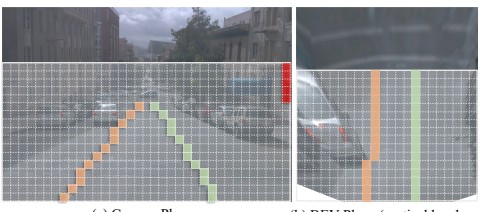

(a) Camera Plane     (b) BEV Plane (vertical head group)

Figure 6: Row-wise classification in the camera plane and BEV plane (vertical head group). The rows where no lanes are visible are marked in red. The grids marked in orange and green are crossed by two different lane predictions, which correspond to different channel groups.

of grids in Fig. 6 (b) contain lanes. Additionally, the lanes in the BEV plane are more straight and in parallel with each other. These issues alleviate the optimization difficulty. Given these observations, we argue that row-wise classification naturally favors BEV based 3D lane detection. By applying a softmax operation to the last dimension of the feature map, the output is $y_r \in \mathbb{R}^{B \times N \times H_b \times W_b}$.

**Category head.** The category head is to distinguish which classes the detected lanes belong to. To this end, the shape of the category head output $y_c$ is $(B, NG, 1, 1)$, where $G$ denotes the total category number. It is implemented by pooling the feature of foreground regions containing lanes.

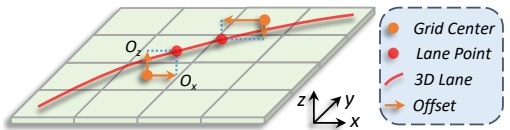

Figure 7: Diagram of the x-axis (or y-axis) and z-axis offsets for the vertical head group.

**Offset heads.** Based on the result from the visibility head and row index head, we can know which BEV grids are crossed by lanes. The z-axis coordinates of all the BEV grid centers are set to 0. However, simply using these grid centers to

Table 1: Performance comparison in the OpenLane validation set. The $1_{st} \sim 4_{th}$ rows of results are previously published methods. The $5_{th} \sim 8_{th}$ rows are concurrent works and only preprinted. The results in the $8_{th} \sim 10_{th}$ rows correspond to GroupLane with ResNet18, ResNet50, and ConvNext-B as the backbones, respectively. The best performance given the metric F1 score is marked in red.

| Method | F1 score (%)↑ | Category Accuracy (%)↑ | X error near↓ | X error far↓ | Z error near↓ | Z error far↓ |
|---|---|---|---|---|---|---|
| Gen-LaneNet (Guo et al., 2020) | 32.3 | - | 0.591 | 0.684 | 0.411 | 0.521 |
| Cond-IPM (Liu et al., 2021a) | 36.3 | - | 0.563 | 1.080 | 0.421 | 0.892 |
| 3D-LaneNet (Garnett et al., 2019) | 44.1 | - | 0.479 | 0.572 | 0.367 | 0.443 |
| PersFormer (Chen et al., 2022) | 50.5 | 92.3 | 0.485 | 0.553 | 0.364 | 0.431 |
| CurveFormer (Bai et al., 2022) | 50.5 | - | 0.340 | 0.772 | 0.207 | 0.651 |
| Anchor3DLane (Huang et al., 2023) | 53.7 | 90.9 | 0.276 | 0.311 | 0.107 | 0.138 |
| BEV-LaneDet (Huang et al., 2023) | 58.4 | - | 0.309 | 0.659 | 0.244 | 0.631 |
| PETRv2 Liu et al. (2023) | 61.2 | - | - | - | - | - |
| LATR Luo et al. (2023) | 61.9 | 92.0 | 0.219 | 0.259 | 0.075 | 0.104 |
| GroupLane-Res18 | 55.4 | 90.9 | 0.441 | 0.483 | 0.262 | 0.354 |
| GroupLane-Res50 | 60.2 | 91.6 | 0.371 | 0.476 | 0.220 | 0.357 |
| GroupLane-CN-B | 64.1 | 92.8 | 0.320 | 0.441 | 0.233 | 0.402 |

Table 2: Performance comparison in the Once-3DLanes validation set. The 1st~4th rows of results are previously published methods. The 5th row corresponds to a concurrent work and is only preprinted. The results in the 6th~8th rows are GroupLane with ResNet18, ResNet50, and ConvNext-B as the backbones, respectively. The best performance given the primary metric F1 score is marked in red.

| Method | F1 score (%)↑ | Precision(%)↑ | Recall (%)↑ | CD error↓ |
|---|---|---|---|---|
| 3D-LaneNet (Garnett et al., 2019) | 44.73 | 61.46 | 35.16 | 0.127 |
| Gen-LaneNet (Guo et al., 2020) | 45.59 | 63.95 | 35.42 | 0.121 |
| SALAD (Yan et al., 2022) | 64.07 | 75.90 | 55.42 | 0.098 |
| PersFormer (Chen et al., 2022) | 74.33 | 80.30 | 69.18 | 0.074 |
| Anchor3DLane (Huang et al., 2023) | 74.87 | 80.85 | 69.71 | 0.060 |
| GroupLane-Res18 | 80.73 | 82.56 | 78.90 | 0.053 |
| GroupLane-Res50 | 79.69 | 82.25 | 77.29 | 0.055 |
| GroupLane-CN-B | 79.42 | 82.41 | 76.54 | 0.054 |

describe the positions of these 3D lanes is imprecise. Therefore, we regress the x-axis and z-axis offsets for the vertical head group to refine the results. The diagram of the x-axis and z-axis offsets is illustrated in Fig. 7. Similarly, for the horizontal head group, we predict the y-axis and z-axis offsets relative to BEV grid centers.

# 4 EXPERIMENTS

## 4.1 EXPERIMENTAL DETAILS

For the experiments in Section 4.3, we report the results of GroupLane using various backbones, i.e., ResNet-18, ResNet-50, and ConvNext-Base. In the remaining experiments, we only report the results of GroupLane with ResNet-50 as the backbone. All the backbones are pre-trained on ImageNet (Deng et al., 2009) and no extra data is adopted. The neck downsamples the input feature map resolution to $\frac{1}{16}$ of the originally captured image. The BEV grid resolution is set to $24 \times 100$. The channel group number $N$ is set to 16. The BEV encoder consists of 4 convolutional blocks and an FPN. All the experiments are conducted on 8 RTX2080 GPUs. No tricks like model ensenmble and test-time augmentation are used. In all evaluation datasets, the detector is trained for 10 epochs using the Adamw optimizer (Loshchilov & Hutter, 2017), and the learning rate is set to $2e - 4$.

## 4.2 BENCHMARKS

**OpenLane.** OpenLane is a real-world 3D lane detection dataset built on top of Waymo (Sun et al., 2020), a large-scale autonomous driving dataset. About 200K frames of images captured by a front-view camera are included in OpenLane. In these images, over 880K 3D lane instances are annotated. Besides localizing the lanes, OpenLane also requires the evaluated detectors to identify the categories of lanes. There are totally 14 lane categories in OpenLane. The employed evaluation metrics include the F1 score, category accuracy, X error near, X error far, Z error near, and Z error far. Among them, F1 score is the most important metric.

Table 3: Efficiency comparison between GroupLane and PersFormer.

| Method | Backbone | IS (FPS)↑ | FLOPs ($\times 10^9$)↓ | F1 score (%)↑ |
|---|---|---|---|---|
| PersFormer | EfficienNet-B7 | 5.58 | 572.4 | 50.5 |
| GroupLane-Res18 | ResNet18 | 38.17 | 76.4 | 55.4 |
| GroupLane-Res50 | ResNet50 | 19.84 | 146.1 | 60.2 |
| GroupLane-CN-B | ConvNext-Base | 13.23 | 233.2 | 64.1 |

**Once-3DLanes.** Once-3DLanes is another large-scale real-world 3D lane detection dataset labeled based on Once (Mao et al., 2021). About 211K frames of images captured by a front-view camera are included in this dataset. Compared with OpenLane, it only demands evaluated detectors to localize the lanes but not classify them. The considered evaluation metrics include the F1 score, Precision, Recall, and CD error. Among them, the F1 score plays the most critical role.

**OpenLane-Huawei.** Different from OpenLane and Once-3DLanes that require detectors to recognize 3D lanes, OpenLane-Huawei demand detectors to detect 3D lane centerlines. The OpenLane-Huawei dataset includes 1000 scenes of videos with roughly 15s duration and the videos are captured by 7 cameras facing in different directions. About 960K instance annotations are contained. The evaluation metrics are the F1 score, Recall, Precision, and DET-L score. Compared with OpenLane and Once-3DLanes that almost do not contain horizontal lanes, there exist numerous horizontal lane centerlines in OpenLane-Huawei. Hence, OpenLane-Huawei is more suitable for evaluating the horizontal lane detection capability of GroupLane.

## 4.3 COMPARISON WITH PREVIOUS SOTAS

**OpenLane.** We train GroupLane with various backbones (ResNet18, ResNet50, and ConvNext-Base) using the training set of OpenLane. The evaluation results of GroupLane and its compared methods in the OpenLane validation set are presented in Table 1[1]. The top 4 rows are published methods and the remaining ones are our concurrent works.

As presented, GroupLane outperforms all compared methods by large margins and establishes new SOTA. It ranks the first in both the settings of being compared with BEV counterparts and all previous methods. For example, GroupLane-CN-B surpasses PersFormer, by 13.6% F1 score. According to the category accuracy results, GroupLane also obtains the highest classification precision. In addition, we can observe that GroupLane using a bigger backbone like ConvNext-Base outperforms the one with a naive backbone (such as ResNet18) in OpenLane.

**Once-3DLanes.** In this experiment, all methods are trained with the Once-3DLanes training set and evaluated in the validation set. The results are reported in Table 2. According to the results, GroupLane behaves the best among all the methods. For instance, GroupLane-Res18 outperforms PersFormer, the best one among published detectors, by 6.40% F1 score. This result further confirms the superiority of GroupLane.

Notably, an interesting observation found in Table 2 is that GroupLane using a smaller backbone achieves slightly better performance. For example, GroupLane-Res18 surpasses GroupLane-CN-B by 1.31% F1 score. We speculate that this is because there is no need to classify lanes in Once-3DLanes. Additionally, the lanes are white or yellow curves, which are easy to identify. Therefore, a small backbone like ResNet18 is strong enough to produce discriminative representation in Once-3DLanes. Besides, backbones with fewer parameters are easier for optimization. Thus, GroupLane with ResNet18 as the backbone behaves better than the one using ConvNext-Base.

**Efficiency Comparison.** 3D lane detection is a task needing to be applied to real vehicles, and thus efficiency is important. However, previous literature mostly only compares metrics about detection precision and hardly studies this efficiency issue. In this part, we hope to bridge this gap. To this end, we compare GroupLane with PersFormer, in the metrics of inference speed (IS) and FLOPs. The F1 score results are also given to reflect the detection precision under different computing costs. The experiment is conducted using the OpenLane benchmark, and the results are presented in Table 3.

As shown, PersFormer adopts EfficientNet-B7 as the backbone, the computing cost of which is much heavier than ours. We do not report the result of GroupLane using EfficientNet-B7 because we find RTX2080 cannot support its training. In Table 3, we report the performance of GroupLane with

---

[1]The listed methods for comparison are up to November 2023.

Table 4: Performance comparison between GroupLane with and without the horizontal head group. This experiment is performed using the OpenLane-Huawei benchmark. The 1st row of results corresponds to GroupLane without using the horizontal head group, and the 2nd row of results is the GroupLane using them.

| H-SOM | F1 score (%)↑ | Recall (%)↑ | Precision (%)↑ | **DEL-L**↑ |
|---|---|---|---|---|
| No | 31.00 | 27.51 | 35.51 | 9.34 |
| Yes | 35.82 | 32.67 | 39.64 | 17.07 |

Table 5: Ablation study on the effect of the channel grouping strategy. The first column indicates whether to use the group convolution to decouple feature interaction. The second column is the convolutional channel number for each group.

| Group | Num | **F1 score** (%)↑ | Category accuracy (%)↑ | X error near↓ | X error far↓ |
|---|---|---|---|---|---|
|  | 4 | 57.27 | 90.17 | 0.390 | 0.495 |
| ✓ | 4 | 57.70 | 89.94 | 0.417 | 0.489 |
| ✓ | 8 | 58.27 | 91.21 | 0.397 | 0.497 |
| ✓ | 16 | 60.24 | 91.61 | 0.371 | 0.476 |
| ✓ | 32 | 59.73 | 91.45 | 0.368 | 0.452 |

three different backbones, i.e., ResNet18, ResNet50, and ConvNext-Base. We can observe that even GroupLane with ResNet18 still outperforms PersFormer using EfficientNet-B7 by 4.9% F1 score, while the inference speed is nearly 7× faster than it. Meanwhile, the FLOPs of GroupLane-Res18 is only 13.35% of PersFormer. All these observations suggest that GroupLane is a very efficient 3D lane detector and more friendly to practical applications.

## 4.4 STUDY ON TRAINING DYNAMICS

In this experiment, we study the training dynamics of GroupLane. To this end, we train GroupLane with the OpenLane training set for 10 epochs and evaluate its performance per two epochs. The F1 score and classification accuracy dynamics of GroupLane are illustrated in Fig. 8.

As shown, GroupLane converges well at about the 4th epoch, which indicates its convergence is very fast. In fact, taking ResNet50 as the backbone, the training process of GroupLane only takes about 12 hours (8 RTX2080 GPUs, the batch size is 16). By contrast, PersFormer needs 100 training epochs. This observation suggests that GroupLane not only achieves superior performance in the inference process, it is also very economical for the training phase due to its fast convergence speed.

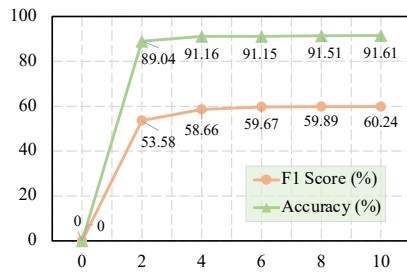

Figure 8: The training dynamics of GroupLane in OpenLane. The F1 score and classification accuracy curves are obtained via evaluating GroupLane using the OpenLane validation set.

## 4.5 ABLATION STUDY

This part ablates the effectiveness of the horizontal head group and channel grouping strategy. In addition, two association strategies are compared.

**Horizontal Head Group.** Both OpenLane and Once-3DLanes are not suitable to verify the effectivess of the horizontal head group because their data contains few horizontal lanes. Thus, we employ OpenLane-Huawei to bridge this gap. Compared with OpenLane and Once-3DLanes, there are numerous crossroad scenes and horizontal lane centerline instances in OpenLane-Huawei, which sufficiently reflects the horizontal lane detection capability of GroupLane. The results are reported in Table 4. The input image resolution is (640, 480) and DET-L is the primary metric. As presented, the horizontal head group almost doubles the detection performance of GroupLane, which suggests the effectiveness of our design.

**Channel Grouping Strategy.** In this experiment, we validate the effectiveness of channel grouping. On the one hand, we study the influence of splitting features into various groups based on group convolution. On the other hand, we also analyze how the channel number for each group of features affects the detection performance. The results of this experiment are given in Table 5.

Two observations are drawn from Table 5. First of all, decoupling the feature interaction between different groups boosts the F1 score slightly (0.51% F1 score). We attribute this improvement to the

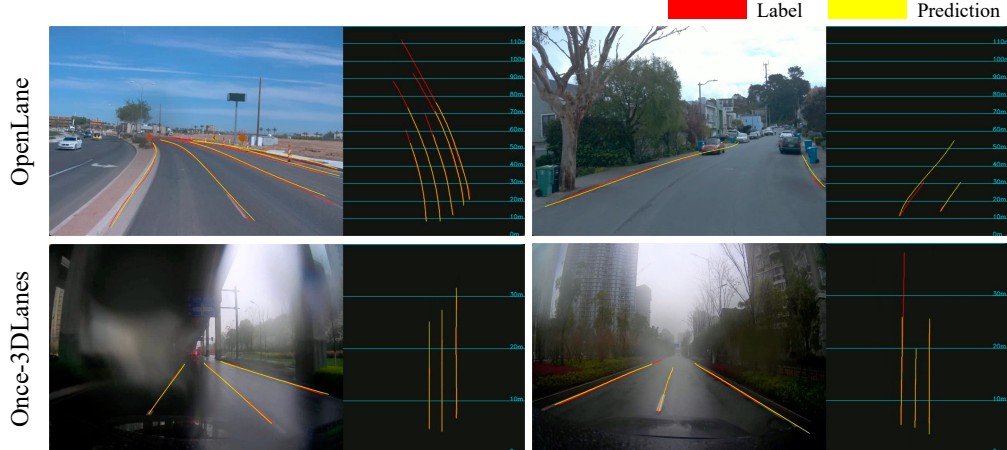

Figure 9: Detection result examples of GroupLane in the OpenLane and Once-3DLanes validation sets. The red and yellow lanes indicate ground truth and prediction, respectively.

alleviation of optimization difficulty. Secondly, increasing the convolution channel number for every feature group first improves the performance and then decreases the performance. We infer that the feature group channel number behaves similarly to the embedding length of the Transformer query. Only a few channel number causes the network not strong enough to capture the representation of objects, and numerous channels increase the optimization difficulty significantly.

**Comparing Association Strategies.** GroupLane adopts the developed SOM to match predictions with labels, which leads to impressive performance. Differently, previous methods following the row-wise classification paradigm usually associate predictions and targets by simply matching them with the same indexes. We argue this strategy is suboptimal and improper for realizing end-to-end 3D lane detection, because it does not match a prediction with its most similar target. Therefore, we use SOM. In this experiment, we compare the performances of GroupLane using these two different matching strategies, and the results are reported in Table 6. The results suggest that the F1 score of matching based on indexes is only 40.9% relatively of the one using SOM.

Table 6: Ablation study on the association between predictions and lane labels based on SOM.

| SOM | F1 score (%)↑ | Category accuracy (%)↑ | X error near↓ | X error far↓ |
|---|---|---|---|---|
| No | 24.5 | 65.1 | 1.36 | 1.05 |
| Yes | 60.2 | 91.6 | 0.371 | 0.476 |

### 4.6 VISUALIZATION AND LIMITATION

Some result examples of GroupLane are presented in Fig. 9. As shown, the detection precision of GroupLane in the BEV plane is promising. The prediction deviation in the camera image plane is caused by the depth estimation error, which is unavoidable because monocular depth estimation is an ill-posed task. This problem may be addressed by introducing depth information from lidar points, and we plan to study this issue in our future work.

### 4.7 ETHICAL STATEMENT

The proposed algorithm is an efficient detector for 3D lane detection, which is an important and widely studied task used in autonomous driving. In this work, we do not think the developed detector has potential ethical risk needing special discussion.

## 5 CONCLUSION

In this work, we have proposed a novel 3D lane detector, GroupLane, that realizes precise, fast, and end-to-end 3D lane detection. Through dividing convolutional feature into multiple groups in the channel dimension and employ every group to represent a prediction instance, GroupLane enjoys the end-to-end characteristic like DETR while maintaining the small computational cost of CNN. Besides, we have proposed to perform row-wise classification in BEV and devised a set of detection heads. This design further boosts the detection precision. Extensive experiments have been conducted on various benchmarks to demonstrate the superiority of GroupLane.

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

CONTENTS

# A APPENDIX

We present more details about our method here due to the 9-page text limitation.

## A.1 VISUALIZATION ON OPENLANE-HUAWEI

Due to the 9-page text limitation, we do not visualize detection results on OpenLane-Huawei in Fig. 9. Inseadly, we present them in Fig. 10, where both the results of detectors without and with the devised horizontal head group are shown. As illustrated, GroupLane without the horizontal head group fails to recognize these horizontal lane centerlines. By contrast, GroupLane with the horizontal head group detects them successfully. Besides, we can observe that the vertical lane detection results of GroupLane without the horizontal head group is significantly more imprecise than the one with the horizontal head group. We believe this is because when a detector without detecting horizontal lane capability is forced to learn recognizing horizontal lanes, the stability of normal training process is disturbed.

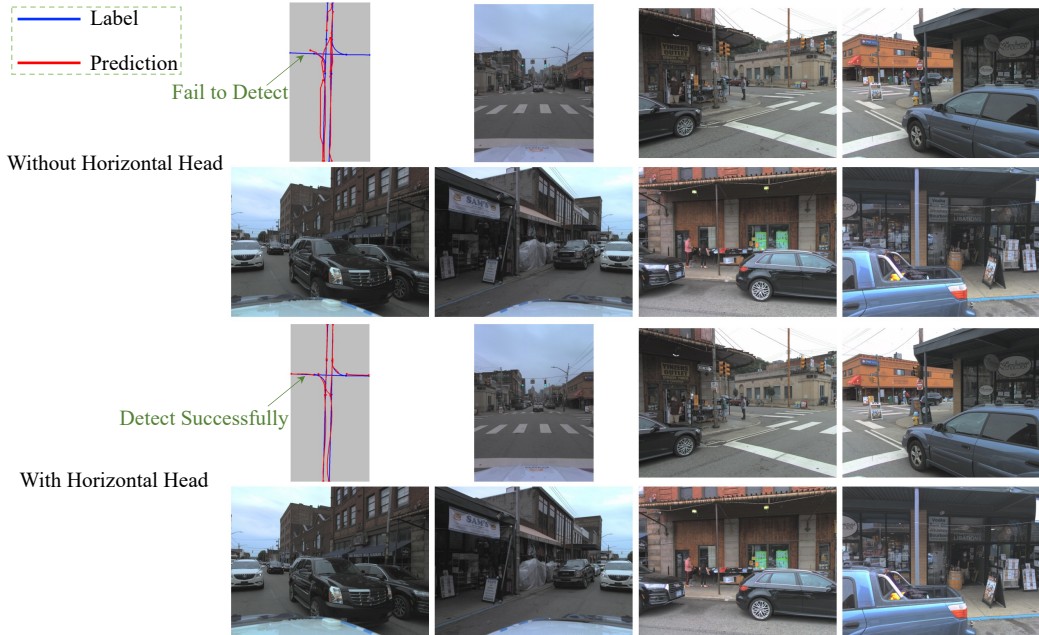

Figure 10: Visualization of detection results with and without using the designed horizontal head group in OpenLane-Huawei.

## A.2 DISCUSSION ON THE EFFICIENCY OF GROUPLANE

Although very simple, GroupLane outperforms previous counterparts by large margins on detection precision. Besides, its inference speed and convergence dynamics are also impressive. Therefore, in this part, we want to further discuss why GroupLane is so efficient and what is the key of designing 3D lane detectors. There are two issues:

**(I) Producing BEV feature.** The first issue is whether we need to produce the birds'-eye-view (BEV) feature. Although GroupLane generates detection results based on the BEV feature, there are also some works obtain somewhat promising performance without using BEV feature (Huang et al., 2023). We believe the BEV feature generation step is necessary. One the one hand, we think this step is beneficial to the 3D lane detection precision, because the transformation from camera view feature to the BEV feature exactly helps matching the information in the camera view coordinate system and BEV coordinate system. Since the image feature is in the camera view coordinate system and the detection results need to be in the BEV coordinate system, this matching process contributes to better performance. On the other hand, generating BEV feature is also beneficial to future development of 3D lane detection. Current 3D lane detection is restricted to the front camera view, because no

multi-view 3D lane detection dataset is available until now. However, when a multi-view 3D lane detection dataset is available, the 3D lane detection framework based on BEV feature can naturally tackle it. Besides, this framework can be extented to support multiple kinds of sensors and various downstream tasks flexibly (Chen et al., 2023; Zhang et al., 2022).

**(II) Trade-off between feature extraction ability and optimization difficulty.** Secondly, we speculate that there are two issues primarily influencing the performance of a model on a task, i.e., the feature extraction ability and optimization difficulty. For example, with the growth of model parameter volume, a model presents stronger feature extraction ability but the optimization difficulty is also enhanced. To tackle the optimization difficulty, more training data and better optimization strategy are demanded. Besides, the optimization difficulty can also be alleviated by designing a model architecture easier for optimization. Since the publicly available training data volume for 3D lane detection is limited, designing a better model architecture is crucial.

As mentioned in the paper, there exist mainly 4 lane detection paradigms, i.e., segmentation based, anchor based, parametric curve based, and row-wise classification based. Among them, the segmentation based paradigm needs to build an embedding head to associate segmented foreground grids as lanes, which is a very challenging task. Hence, we do not use this paradigm. The performance of anchor based methods rely heavily on the manual design of anchors, which is thus unsuitable. The parametric curve based paradigm directly regresses the parameters of parametric curves, which is end-to-end but quite difficult to learn when training data is limited. Given the aforementioned considerations, we select the row-wise classification based paradigm, because it can realize end-to-end detection but do not show the great optimization difficulty like the parametric curve based paradigm.

Combining all the above insights, we propose GroupLane, which first produces the BEV feature and then conducts row-wise classification on the BEV feature. Our extensive experiments shown in the paper confirm the efficiency of this detection architecture.

### A.3 CODE AND REPRODUCIBILITY

Our code is implemented based on a private code framework that is not publically available now. The basic architecture of this code framework is similar to Pytorch-Lightning. To help readers reproduce our work, we have uploaded the core implementation code of GroupLane as the supplementary material, and we will release the implementation based on a public code framework later.

### A.4 MORE DETAILS OF DETECTION HEADS

In this part, we present more implementation details of the detection heads in GroupLane, which are presented as follows.

**Existence head.** As mentioned in Section 3.3, $2N$ detection candidates are generated by the horizontal and vertical head groups, and not all candidates are valid targets. Some candidates correspond to background regions. The existence head is to predict the confidence that a candidate is a valid lane rather than belonging to background regions.

Taking $F_b^v$ or $F_b^h$ as input and processing each of them with a global max pooling layer and a Sigmoid layer, the obtained feature shape is $y_e \in \mathbb{R}^{B \times N}$. Denoting the corresponding label as $\bar{y}_e \in \mathbb{R}^{B \times N}$, the existence head loss $L_e$ is formulated as:

$$L_e = -\frac{1}{N_l} \sum_{i=1}^{BN} [y_e^i \log \bar{y}_e^i + (1 - y_e^i) \log(1 - \bar{y}_e^i)], \qquad (1)$$

where $y_e^i$ and $\bar{y}_e^i$ represent the $i_{\text{th}}$ elements in $y_e$ and $\bar{y}_e$. $N_l$ denotes the number of valid lanes.

**Visibility head.** In this work, we split the BEV plane into uniform grids. The shape of these grids is $H_b \times W_b$, which is the same as the BEV feature resolution. Then, for the vertical head group, a set of y-axis lines ($\{y = y_i\}_{i=1}^{H_b}$) are pre-defined in this BEV space (x-axis lines for the horizontal head group). The visibility head is to estimate whether there are visible lanes crossing these y-axis lines.

To predict whether a row is crossed by a lane correctly, the output of the visibility head needs to aggregate the information from all grids in this row. To this end, the visibility head applies the max

pooling operation to the input feature along the x-axis dimension. The output can be represented as $y_v \in \mathbb{R}^{B \times N \times H_b \times 1}$, and the loss is:

$$L_v = -\frac{1}{N_l H_b} \sum_{i=1}^{BNH_b} y_v^i \log \bar{y}_v^i, \tag{2}$$

where $\bar{y}_v \in \mathbb{R}^{B \times N \times H_b \times 1}$ denotes the corresponding label.

**Row index head.** The row index head applies a softmax operation to the last dimension of the feature map for producing the output $y_r \in \mathbb{R}^{B \times N \times H_b \times W_b}$. The grid corresponding to the highest confidence in every row of $y_r$ is the predicted grid crossed by lanes. The loss $L_r$ is computed as:

$$L_r = -\frac{1}{N_l} \sum_{i=1}^{BNH_b} [\bar{y}_v^i \sum_{j=1}^{W_b} y_r^{(i,j)} \log \bar{y}_r^{(i,j)}], \tag{3}$$

where $\bar{y}_r \in \mathbb{R}^{B \times N \times H_b \times W_b}$ represents the label. $\bar{y}_v^i$ ensures that the loss is computed only using the rows with lanes.

**Category head.** The shape of the category head output $y_c$ is $(B, NG, 1, 1)$, where $G$ denotes the total category number. The row index head output $y_r$ can be regarded as a BEV instance segmentation map, where each channel corresponds to a lane instance. Therefore, we can use $y_r$ to obtain the foreground regions of features in $F_b$, which is denoted as $F_f \in \mathbb{R}^{B \times NG \times H_b \times 1}$. Afterwards, an MLP layer is applied to $F_f$ to aggregate the information in the height dimension, and the output is denoted as $y_c \in \mathbb{R}^{B \times NG \times 1 \times 1}$. The loss $L_c$ is calculated as:

$$L_c = -\frac{1}{N_l} \sum_{i=1}^{BNG} y_c^i \log \bar{y}_c^i, \tag{4}$$

where $\bar{y}_c \in \mathbb{R}^{B \times NG \times 1 \times 1}$ is the corresponding label.

**Offset heads.** To describe the location of this 3D lane precisely, we need to estimate the offsets from the grid centers to the lane points. To this end, we build the x-axis head and z-axis head to regress the offsets. By taking out the foreground regions of head outputs like the category head, the outputs of the x-axis head and z-axis head can be denoted as $y_x \in \mathbb{R}^{B \times N \times H_b \times 1}$ and $y_z \in \mathbb{R}^{B \times N \times H_b \times 1}$. The loss of these two heads $L_o$ are computed as:

$$L_o = -\frac{1}{N_l} [ \sum_{i=1}^{BNH_b} \bar{y}_v^i |y_x^i - \bar{y}_x^i|) + \sum_{i=1}^{BNH_b} \bar{y}_v^i |y_z^i - \bar{y}_z^i|)], \tag{5}$$

where $\bar{y}_x \in \mathbb{R}^{B \times N \times H_b \times 1}$ and $\bar{y}_z \in \mathbb{R}^{B \times N \times H_b \times 1}$ denote the labels of the x-axis and z-axis offsets.

The total loss $L$ is the sum of the above losses of different detection heads:

$$L = L_e + L_v + L_r + L_c + L_o. \tag{6}$$

The matching cost is computed in the same way as the loss function.

### A.5 PERFORMANCE INCONSISTENCY BETWEEN OPENLANE AND ONCE-3DLANES

According to the results in Table 1 and Table 2, the influence of the backbone parameter volume to the performance on the OpenLane and Once-3DLanes dataset is inconsistent. Specifically, larger backbones boost the performance in OpenLane while decreasing the performance in Once-3DLanes. In this part, we delve into this issue.

Specifically, we hypothesize the reason that using ResNet18 behaves better than ResNet50 in the Once-3DLanes dataset is that Once-3DLanes does not require classifying lanes and only localization is needed. By contrast, OpenLane requires both classification and localization. Therefore, the main point causing the inconsistency between Table 1 and Table 2 is that increasing backbone parameter volume primarily benefits classification rather than localization.

Table 7: The influence of backbone selection on the performance in OpenLane when no classification is needed.

| Backbone | ResNet18 | ResNet50 | ConvNext-Base |
|---|---|---|---|
| F1 Score | 71.8% | 71.7% | 71.2% |

To verify this speculation, we remove the classification information in OpenLane. Specifically, we change all the lane category labels in OpenLane as *zero*. Since all lanes share the same class ID, no classification is needed for the detector. In this experimental setting, we validate the performances of GroupLane taking different backbones and the results are reported in Table 7.

As presented in Table 7, our hypothesis holds. In addition, We hypothesize the deeper reason is that: With more parameters in a backbone, better classification capability is obtained from the ImageNet pre-training, which is also a classification task. However, it does not boost the localization ability.

## A.6 More In-Depth Analysis in Head Groups

Although we have studied the influence of the horizontal head group in Table 4, only the overall detection performance is reported due to the page limit. Hereby, we give a more comprehensive analysis. Specifically, we conduct separate evaluations for horizontal and vertical lane detection performances in GroupLane. For the evaluation of horizontal lane detection, we remove the vertical lane labels and predictions. Similarly, for the evaluation of vertical lane detection, we remove the horizontal lane labels and predictions. The results are presented in Table

Table 8: Detailed analysis on the influence of the horizontal head.

| H-SOM | Test Mode | F1 Score | Recall | Precision | DET-L |
|---|---|---|---|---|---|
| No | Vertical | 43.29% | 40.92% | 45.96% | 23.49% |
| No | Horizontal | 19.71% | 16.07% | 25.47% | 1.01% |
| No | Overall | 31.00% | 27.51% | 35.51% | 9.34% |
| Yes | Vertical | 43.10% | 40.66% | 45.86% | 22.01% |
| Yes | Horizontal | 30.11% | 27.81% | 32.82% | 10.94% |
| Yes | Overall | 35.82% | 32.67% | 39.64% | 17.07% |

As observed from the results, the performance of vertical lane detection is higher compared to that of horizontal lanes. This difference can be attributed to the fact that the dataset contains a larger number of vertical lanes compared to horizontal lanes. The imbalanced distribution of lane types in the dataset influences the performance metrics.

## A.7 Why We Choose PersFormer for Comparison in Fig. 2

We select PersFormer to compare for two reasons: (1) This work was completed before the ICCV2023 conference. PersFormer is the published SOTA when we completed this work. (2) Both GroupLane and PersFormer produce BEV feature explicitly, while the current SOTA method LATR is not BEV based.

