# OpenReview forum: "Grouplane: End-to-End 3D Lane Detection with Channel-Wise Grouping"
_ICLR.cc/2024/Conference — Submitted to ICLR 2024_

### Official Review · Reviewer_5Kts · 2023-10-23

**Soundness:** 3 good
**Presentation:** 3 good
**Contribution:** 2 fair
**Rating:** 5
**Confidence:** 4

**Summary:**

The paper presents a novel convolutional neural network for end-to-end 3D lane detection. First, the approach splits BEV features into vertical and horizontal groups along the channel dimension. Then, group convolution is applied to both vertical and horizontal groups to extract further features. During the training phase, the authors propose a SOM strategy to match ground truth and predictions. Finally, 3D lanes are detected by performing row-wise classification. Notably, this method achieves state-of-the-art performance in the 3D lane detection task on 3 benchmarks, OpenLane, Once-3DLanes, and OpenLane-Huawei.

**Strengths:**

1. Idea seems fundamentally sound.
2. Spliting BEV feature for vertical and horizontal lane detection would be valuable, espeically when model is deployed on an edge device.
3. Paper is well written and very easy to read.

**Weaknesses:**

1. Simply dividing each group into N outputs limited the max output lane number of the model.
2. SOM strategy is simple yet effect, details are not well explained or even missing, eg. the matching cost definition.

**Questions:**

1. It is unclear what will happen when both the vertical and horizontal heads match the ground truth (GT). The paper does not provide a clear explanation or analysis of this scenario.
2. The paper does not address how to ensure stable predictions. It raises concerns about the possibility of different heads predicting the same lane at different time and one of them is not the optimal prediction.
3. Can not find the matching cost definition or loss functions in Section 3.4.

---

> ### Author Response · Authors · 2023-11-17
> **Response to Reviewer 5Kts**
>
> We thank the reviewer for the effort and valuable feedback.
>
> **Q1: Simply dividing each group into N outputs limited the max output lane number of the model.**
>
> A1: Firstly, it is important to note that the number of lanes on real roads is typically limited. In most cases, there are at most tens of lanes on a road. Therefore, having an upper bound on the prediction number does not significantly impact the practical deployment of GroupLane in real-world applications.
>
> &ensp;&ensp; Secondly, it is worth mentioning that almost all current lane detectors also have an upper bound on the prediction number. For example, the maximum prediction number of DETR is determined by the number of queries, while anchor-based detectors have a maximum prediction number based on the total number of anchors.
>
> &ensp;&ensp; In comparison to other algorithms, GroupLane offers several advantages, including being more lightweight, faster, and demonstrating better precision. Consequently, it is relatively straightforward to increase the maximum prediction number of GroupLane beyond that of other detectors if necessary.
>
> &ensp;&ensp; Considering the aforementioned points, we firmly believe that having a maximum prediction number does not undermine the superiority of GroupLane. The model's performance and efficiency, combined with the practical limitations of lane numbers on real roads, make GroupLane a highly effective solution for lane detection tasks.
>
> **Q2: SOM strategy is simple yet effect, details are not well explained or even missing, eg.
> the matching cost definition.**
>
> A2: We apologize for the oversight in not providing sufficient details on the SOM strategy in the paper. We have provided a more comprehensive explanation of the matching cost definition and other relevant details in Appendix Section A.4 of the paper, which has been highlighted in cyan. We do not present it here because it is too long and the reply box has a character limit. We hope the reply can address the concern of the Reviewer well.
>
> **Q3: It is unclear what will happen when both the vertical and horizontal heads match the ground truth (GT). The paper does not provide a clear explanation or analysis of this scenario.**
>
> A3: It seems that there are some misunderstanding about our method. In fact, the single-win one-to-one matching strategy employed in our approach explicitly matches one GT with only one prediction from the vertical and horizontal heads (details can refer to Section 3.3 in the paper). This ensures that the scenario where both heads match the same GT does not exist in our framework.
>
>
> **Q4: The paper does not address how to ensure stable predictions. It raises concerns about the possibility of different heads predicting the same lane at different time and one of them is not the optimal prediction.**
>
> A4: The issue of unstable predictions is not a concern in GroupLane. The matching process employed in GroupLane is based on the same principles as the DETR algorithm [1]. The one-to-one matching algorithm ensures that only the predictions that are most similar to the ground truth (GT) are matched. Any predictions that do not match any GT are treated as background.
>
> If a matched prediction for a lane is not of sufficient quality, the loss function will penalize the prediction and encourage the model to generate a better one. This mechanism ensures that the algorithm remains stable by continuously refining and improving the predictions.
>
> [1] Carion N, Massa F, Synnaeve G, et al. End-to-end object detection with transformers[C]//European conference on computer vision.  2020: 213-229.

---

> > ### Author Response · Authors · 2023-11-21
> > **Message to the Reviewer 5Kts**
> >
> > Dear Reviewer 5Kts,
> >
> > We are sincerely grateful to you for the precious time and selfless efforts you have devoted to reviewing our paper.
> >
> > We have provided our detailed response to each concern. Since the deadline for reviewer-author discussion is approaching, we would like to inquire whether our response has addressed your concerns and if you have the time to provide further feedback on our rebuttal. We are more than willing to engage in further discussion.
> >
> > Best regards,
> >
> > The Authors

---

### Official Review · Reviewer_dGoq · 2023-10-30

**Soundness:** 2 fair
**Presentation:** 3 good
**Contribution:** 3 good
**Rating:** 6
**Confidence:** 3

**Summary:**

This paper presents a end-to-end 3D lane detection from a single image. The proposed model is based on technical contributions: (1) a splitting strategy that build several groups of features to represent a line, (2) two groups of heads to recognize, in the bird-eye-view (BEV), horizontal and vertical lines. The resulting model is evaluated on three public benchmarks and outperform existing models.

**Strengths:**

The paper is nicely written and easy to read. The main contribution, to my point of view, consists in splitting the BEV features into two groups of candidates: horizontal candidates and vertical candidates. Each group has 6 heads to predict existence confidence, visibility, category, row-wise classification index, x-axis offset, and z-axis offset. Since the proposed model splits the group of candidates in horizontal and vertical, the authors proposed an adapted technic called single-win one-to-one matching (SOM) to match each candidate with the training labels.

**Weaknesses:**

In the experimental part, GROUPLANE is evaluated on three datasets. The selected baseline model is PersFormer (described as the best published model). Can you give details on this choice? Regarding the benchmark webpage, it seems that the best 2022 model is 58% F1 score and that PersFormer is currently ranked 9. The resulting figure 2 is not fair and should be changed with new models.
Moreover, can you add the two following references (ranked 1 and 2) from ICCV2023:
LATR: 3D Lane Detection from Monocular Images with Transformer
PETRv2: A Unified Framework for 3D Perception from Multi-Camera Images

In the ablation study, the authors compare the Horitontal/Vertical grouping strategy with only a vertical strategy. The proposed strategy increases about 5% the F1 score. It should be interesting to give information on the horizontal/vertical ratio of lines of the dataset. Moreover, it could be interesting to split the results into vertical/horizontal lines.

**Questions:**

Can you give details on the choice of PersFormer as the baseline for figure 2 and table 3?
Can you give information on the horizontal/vertical ratio of lines of the dataset used for table 6?

---

> ### Author Response · Authors · 2023-11-17
> **Response to Reviewer dGoq**
>
> We thank the reviewer for the effort and valuable feedback.
>
> **Q1: In the experimental part, GROUPLANE is evaluated on three datasets. The selected baseline model is PersFormer (described as the best published model). Can you give details on this choice? Regarding the benchmark webpage, it seems that the best 2022 model is 58% F1 score and that PersFormer is currently ranked 9. The resulting figure 2 is not fair and should be changed with new models. Moreover, can you add the two following references (ranked 1 and 2) from ICCV2023: LATR: 3D Lane Detection from Monocular Images with Transformer PETRv2: A Unified Framework for 3D Perception from Multi-Camera Images.**
>
> A1: We apologize for the oversight in not considering more recent publications, as this work was completed before the ICCV2023 conference. We appreciate you bringing this to our attention. We compare our method with PersFormer in Figure 2 of the paper for two reasons: (1) PersFormer is the published SOTA when we completed this work. (2) Both GroupLane and PersFormer produce BEV feature explicitly, while the current SOTA method LATR is not BEV based.
>
>  In the revised version of our paper, we have made the necessary updates. We remove all sentences that claim PersFormer is the published SOTA, and add the reason why we choose PersFormer to compare in Figure 2 to Appendix A.7. The results of PETRv2 and LATR are added to Table 1 (Page 6) of the paper for performance comparison. The references you mentioned are also added (page 2 and Page 3). The updated content is marked in blue.
>
> **Q2: In the ablation study, the authors compare the Horitontal/Vertical grouping strategy with only a vertical strategy. The proposed strategy increases about 5% the F1 score. It should be interesting to give information on the horizontal/vertical ratio of lines of the dataset. Moreover, it could be interesting to split the results into vertical/horizontal lines.**
>
> A2: We apologize for the lack of clarity in the paper. Here, we provided the following details:
>
> In the OpenLaneV2 dataset, approximately 71% of lane instances are vertical lanes, while the remaining 29% are horizontal lanes.
>
> Following your suggestion, we conducted separate evaluations for horizontal and vertical lane detection performances in GroupLane. For the evaluation of horizontal lane detection, we removed the vertical lane labels and predictions. Similarly, for the evaluation of vertical lane detection, we removed the horizontal lane labels and predictions. The results are presented below:
>
>
> | H-SOM | Test Mode | F1 Score| Recall | Precision | DET-L |
> | :-: | :-: | :-: | :-: | :-:| :-:|
> | No | Vertical | 43.29% | 40.92% | 45.96% | 23.49% |
> | No | Horizontal | 19.71% | 16.07% | 25.47% | 1.01% |
> | No | Overall | 31.00% | 27.51% | 35.51% | 9.34% |
> | Yes | Vertical | 43.10% | 40.66% | 45.86% | 22.01% |
> | Yes | Horizontal | 30.11% | 27.81%  | 32.82% | 10.94% |
> | Yes | Overall | 35.82% | 32.67% | 39.64% | 17.07% |
>
> As observed from the results, the performance of horizontal lane detection is higher compared to that of vertical lanes. This difference can be attributed to the fact that the dataset contains a larger number of vertical lanes compared to horizontal lanes. The imbalanced distribution of lane types in the dataset influences the performance metrics.
>
> We have added this experiment analysis to the paper, which is presented in Appendix A.6 and marked in blue.

---

> > ### Author Response · Authors · 2023-11-21
> > **Message to the Reviewer dGoq**
> >
> > Dear Reviewer dGoq,
> >
> > We are sincerely grateful to you for the precious time and selfless efforts you have devoted to reviewing our paper.
> >
> > We have provided our detailed response to each concern. Since the deadline for reviewer-author discussion is approaching, we would like to inquire whether our response has addressed your concerns and if you have the time to provide further feedback on our rebuttal. We are more than willing to engage in further discussion.
> >
> > Best regards,
> >
> > The Authors

---

> > > ### Comment · Reviewer_dGoq · 2023-11-23
> > >
> > > I thank the authors for the answers and additional experiments.
> > > I think there is a mistake in the answer :
> > > "As observed from the results, the performance of horizontal lane detection is higher compared to that of vertical lanes"
> > > should be
> > > "As observed from the results, the performance of VERTICAL lane detection is higher compared to that of HORIZONTAL lanes"
> > >
> > > Regarding PersFormer as SOTA, I appreciate that the sentences have been removed. When comparing your contribution to SOTA on a public dataset with an open competition, you should mention the exact date of the comparison. Moreover, you should mention bot the ranking of your model in the overall models and the ranking of your model in the same class of models (BEV based solutions).
> > >
> > > Best

---

> > > > ### Author Response · Authors · 2023-11-23
> > > > **Response to Reviewer dGoq**
> > > >
> > > > We thank the reviewer for the effort and valuable feedback.
> > > >
> > > > **Q1: I think there is a mistake in the answer: "As observed from the results, the performance of horizontal lane detection is higher compared to that of vertical lanes" should be "As observed from the results, the performance of VERTICAL lane detection is higher compared to that of HORIZONTAL lanes.**
> > > >
> > > > A1: We thank the Reviewer for pointing out our typo. We have corrected it in the paper, which is marked in blue in Section A.6 of the Appendix.
> > > >
> > > > **Q2: Regarding PersFormer as SOTA, I appreciate that the sentences have been removed. When comparing your contribution to SOTA on a public dataset with an open competition, you should mention the exact date of the comparison. Moreover, you should mention both the ranking of your model in the overall models and the ranking of your model in the same class of models (BEV based solutions).**
> > > >
> > > > A2: We thank the Reviewer for this reminder. We have improved the paper as suggested. Specifically, the currently listed methods for comparison in Table 1 of the paper are up to November 2023. We have updated this information in Page 7 of the paper as a footnote, which is highlighted in blue.
> > > >
> > > > In addition, we have also mentioned the rankings of GroupLane compared with overall models and BEV based solutions in Page 7 of the paper and marked in blue. GroupLane ranks first in both settings.

---

### Official Review · Reviewer_RkdN · 2023-10-31

**Soundness:** 3 good
**Presentation:** 2 fair
**Contribution:** 2 fair
**Rating:** 5
**Confidence:** 5

**Summary:**

This paper introduces anchor-based 3D lane detection utilizing channel-wise grouping features. Additionally, the authors propose a single-win one-to-one matching method that associates a grid belonging to vertical or horizontal lanes. The detection heads predict the existence, visibility, row index, lane category, and offset of lane points to grid centers. The paper provides extensive experimental results, demonstrating high performance on various lane detection benchmarks.

**Strengths:**

- Provide test results on various datasets and achieved high performances.

**Weaknesses:**

- The ultra-fast deep lane detection method has already introduced a hybrid anchor-based lane detection that predicts row-and-column anchors corresponding to lanes.
- It is interesting to note that Table 1 and Table 2 exhibit inconsistent results when using different backbone models. It would be nicer if the authors further investigated this issue.

Z. Qin, P. Zhang and X. Li, "Ultra Fast Deep Lane Detection With Hybrid Anchor Driven Ordinal Classification," in IEEE TPAMI, 2022.

**Questions:**

.

---

> ### Author Response · Authors · 2023-11-11
> **Response to Reviewer RkdN**
>
> We thank the reviewer for the effort and valuable feedback.

---

> ### Author Response · Authors · 2023-11-17
> **Response to Reviewer RkdN**
>
> We thank the reviewer for the effort and valuable feedback.
>
> **Q1: The ultra-fast deep lane detection method has already introduced a hybrid anchor-based lane
> detection that predicts row-and-column anchors corresponding to lanes.**
>
> A1: Please kindly note that we had cited the method of UFLD as a reference (in the 2nd page: Qin et al., 2020). To address you concern, we give a deep comparison here.
>
> * Though UFLD proposes row-wise classification lane detection, its application is limited to 2D lane detection, which is significantly different from that in 3D lane detection.
>
> * To successfully apply the row-wise classification lane detection to 3D lane detection, we have conducted multiple improvements:
>
>  &ensp;&ensp; (1) First of all, the row-wise classification lane detection method is conducted on the BEV plane, instead of the camera plane (adopted by UFLD). This is because we consider the geometry of lanes is simpler in the BEV plane and thus easier to learn as shown in Figure 6 of the paper.
>
> &ensp;&ensp; (2) Secondly, we build two groups of heads, one for horizontal lanes detection and the other for vertical lanes detection.
>
> &ensp;&ensp; (3) Thirdly, a novel one-to-one matching algorithm named single-win one-to-one matching (SOM) is developed to match predictions with targets during training for computing loss.
>
> &ensp;&ensp; As a result, although UFLD behaves poorly compared with other counterparts in the 2D lane detection domain, our GroupLane achieves strong performance.
>
> &ensp;&ensp; The above discussion has been included in our paper (page 2 and page 3, marked in brown).
>
> **Q2: It is interesting to note that Table 1 and Table 2 exhibit inconsistent results when using
> different backbone models. It would be nicer if the authors further investigated this issue.**
>
> We thank the Reviewer for the suggestion. We have conducted experimental results and analysis as suggested in the follows to address your concern.
>
> * In the paper, we hypothesize the reason that using ResNet18 behaves better than ResNet50 in the Once-3DLanes dataset is Once-3DLanes does not require classifying lanes and only localization is needed. By contrast, OpenLane requires both classification and localization. Therefore, the main point causing the inconsistency between Table 1 and Table 2 is that **increasing backbone parameter volume primarily benefits classification rather than localization**. To verify this speculation, we remove the classification information in OpenLane. Specifically, we change all the lane category labels in OpenLane as *zero*. Since all lanes share the same class ID, no classification is needed for the detector. In this experimental setting, we validate the performances of GroupLane taking different backbones and the results are as follows.
>
> | Backbone | ResNet18 | ResNet50 | ConvNext-Base |
> | :-: | :-: | :-: | :-: |
> | F1 Score |  71.8%  | 71.7% | 71.2% |
>
> &ensp;&ensp;&ensp; According to the results in the table, a similar performance degradation appearing in OpenLane after removing classification. Thus, our hypothesis holds.
>
> * In addition, We hypothesize the deeper reason is that: with more parameters in a backbone, better classification capability is obtained from the ImageNet pre-training, which is also a classification task. However, it does not boost the localization ability.
>
> The analysis on this issue has also been updated to the Appendix A.5 of the paper and is marked in brown.

---

> > ### Author Response · Authors · 2023-11-21
> > **Message to the Reviewer RkdN**
> >
> > Dear Reviewer RkdN,
> >
> > We are sincerely grateful to you for the precious time and selfless efforts you have devoted to reviewing our paper.
> >
> > We have provided our detailed response to each concern. Since the deadline for reviewer-author discussion is approaching, we would like to inquire whether our response has addressed your concerns and if you have the time to provide further feedback on our rebuttal. We are more than willing to engage in further discussion.
> >
> > Best regards,
> >
> > The Authors

---

> > > ### Comment · Reviewer_RkdN · 2023-11-23
> > >
> > > I read the authors' feedback and the additional experiments clear the uncertainties.

---

> ### Author Response · Authors · 2023-11-23
> **Response to the Reviewer RkdN**
>
> Dear Reviewer RkdN,
>
> Thank you for your valuable feedback on our paper. We are grateful for the time and effort you have put into reviewing our work. We have carefully considered all of your concerns and have made significant revisions to address them.
>
> As presented in the paper, this work proposes a 3D lane detector achieving very promising performance and efficiency. Besides, this is the first time of deploying row-wise classification in 3D lane detection. For the first time, a fully CNN based detector can realize end-to-end detection like DETR. This is also the first time that the detecting horizontal lane problem is considered in lane detection.
>
> We kindly ask if there are any remaining concerns that need to be addressed in order to improve the rating of our paper. We believe that the revisions we have made have significantly improved the quality and contribution of our work. Thank you once again for your time and effort in reviewing our work. We look forward to hearing back from you.
>
> Best regards,
>
> The Authors

---

### Meta-Review · Area_Chair_wd5J · 2023-12-15

**Metareview:**

The paper introduces a novel convolutional neural network for end-to-end 3D lane detection. It splits bird-eye-view (BEV) features into vertical and horizontal groups, applies group convolution, and performs row-wise classification. The method outperforms existing models on three benchmarks and introduces anchor-based 3D lane detection using channel-wise grouping features. The authors also propose a single-win one-to-one matching method, predicting lane existence, visibility, row index, lane category, and offset.

## Strengths

The paper presents a well-written and easy-to-read paper on splitting BEV features into horizontal and vertical candidates for lane detection. The authors propose single-win one-to-one matching (SOM) to match each candidate with the training labels, which is beneficial when deployed on edge devices.

## Weaknesses

The paper discusses a model for lane detection using a simple SOM strategy, but lacks clear explanations or analysis of the matching cost definition. It also raises concerns about stable predictions and the possibility of different heads predicting the same lane at different times. The experimental part evaluates GROUPLANE on three datasets, with PersFormer being the best model. The ablation study compares the Horitontal/Vertical grouping strategy with a vertical strategy, increasing the F1 score by 5%. The authors also mention the introduction of a hybrid anchor-based lane detection method.

**Justification For Why Not Higher Score:**

The paper is well presented and proposes a nice idea, but the authors should add the reviewer concerns before being accepted as it is compared with other submissions.

**Justification For Why Not Lower Score:**

N/A

---

### Decision · Program_Chairs · 2024-01-16

Reject